# Potential of Stroke Imaging Using a New Prototype of Low-Field MRI: A Prospective Direct 0.55 T/1.5 T Scanner Comparison

**DOI:** 10.3390/jcm11102798

**Published:** 2022-05-16

**Authors:** Thilo Rusche, Hanns-Christian Breit, Michael Bach, Jakob Wasserthal, Julian Gehweiler, Sebastian Manneck, Johanna Maria Lieb, Gian Marco De Marchis, Marios Nikos Psychogios, Peter B. Sporns

**Affiliations:** 1Department of Radiology, Clinic of Radiology & Nuclear Medicine, University Hospital Basel, University Basel, 4031 Basel, Switzerland; hanns-christian.breit@usb.ch (H.-C.B.); michael.bach@usb.ch (M.B.); jakob.wasserthal@usb.ch (J.W.); julian.gehweiler@usb.ch (J.G.); smanneck@gmail.com (S.M.); johanna.lieb@usb.ch (J.M.L.); marios.psychogios@usb.ch (M.N.P.); peter.sporns@usb.ch (P.B.S.); 2Department of Neurology, University Hospital Basel, University Basel, 4031 Basel, Switzerland; gian.demarchis@usb.ch; 3Department of Diagnostic and Interventional Neuroradiology, University Medical Center Hamburg-Eppendorf, 20246 Hamburg, Germany

**Keywords:** stroke imaging, low-field MRI, reading study, scanner comparison

## Abstract

Objectives: Ischemic stroke is a leading cause of mortality and acquired disability worldwide and thus plays an enormous health-economic role. Imaging of choice is computed-tomographic (CT) or magnetic resonance imaging (MRI), especially diffusion-weighted (DW) sequences. However, MR imaging is associated with high costs and therefore has a limited availability leading to low-field-MRI techniques increasingly coming into focus. Thus, the aim of our study was to assess the potential of stroke imaging with low-field MRI. Material and Methods: A scanner comparison was performed including 27 patients (17 stroke cohort, 10 control group). For each patient, a brain scan was performed first with a 1.5T scanner and afterwards with a 0.55T scanner. Scan protocols were as identical as possible and optimized. Data analysis was performed in three steps: All DWI/ADC (apparent diffusion coefficient) and FLAIR (fluid attenuated inversion recovery) sequences underwent Likert rating with respect to image impression, resolution, noise, contrast, and diagnostic quality and were evaluated by two radiologists regarding number and localization of DWI and FLAIR lesions in a blinded fashion. Then segmentation of lesion volumes was performed by two other radiologists on DWI/ADC and FLAIR. Results: DWI/ADC lesions could be diagnosed with the same reliability by the most experienced reader in the 0.55T and 1.5T sequences (specificity 100% and sensitivity 92.9%, respectively). False positive findings did not occur. Detection of number/location of FLAIR lesions was mostly equivalent between 0.55T and 1.5T sequences. No significant difference (*p* = 0.789–0.104) for FLAIR resolution and contrast was observed regarding Likert scaling. For DWI/ADC noise, the 0.55T sequences were significantly superior (*p* < 0.026). Otherwise, the 1.5T sequences were significantly superior (*p* < 0.029). There was no significant difference in infarct volume and volume of infarct demarcation between the 0.55T and 1.5T sequences, when detectable. Conclusions: Low-field MRI stroke imaging at 0.55T may not be inferior to scanners with higher field strengths and thus has great potential as a low-cost alternative in future stroke diagnostics. However, there are limitations in the detection of very small infarcts. Further technical developments with follow-up studies must show whether this problem can be solved.

## 1. Introduction

Stroke is the second leading cause of death and a major cause of disability worldwide. Its incidence is increasing because the population is aging [1]. Thus, stroke has an enormous health-economic impact with total costs of approximately EUR 26.6 billion in the European Union (EU) [2] and USD 71.55 billion in the USA [1]. The global incidence in the <65 years age group has increased by approximately 25% [3], particularly affecting younger age groups in low- and middle-income countries [2]. Causes for this increase can partly be attributed to inadequate prevention behavior to reduce risk factors and an insufficient infrastructure [4], especially regarding the availability of stroke imaging and further etiological work-up [5,6]. However, even in high-income countries, the demand for stroke imaging will continue to increase with rising stroke incidences [2]. In particular, MR imaging with diffusion-weighted (DW) sequences will become more important as MRI is superior to CT with a higher sensitivity, especially in the acute phase and for smaller lacunar strokes [6]. On the other hand, costs of MRI are significantly higher than those of CT [7] and metallic implants/cardiac pacemakers may pose contraindications. With this in mind, low-field MRI is increasingly coming into focus, offering MR imaging at a much lower cost but also reducing possible complications with metallic implants [5]. Therefore, the aim of our study was to evaluate the performance of a new prototype of low-field MRI in a prospective cohort of suspected stroke patients and to directly compare the diagnostic value to an established 1.5T MRI.

## 2. Materials and Methods

This prospective study was reviewed and approved by the cantonal (Basel Stadt, Switzerland) ethics committee (*BASEC2021 00166*). All included patients signed an informed consent form.

### 2.1. Patient Selection and Data Acquisition

Patient selection and data acquisition was performed from 1 May 2021 to 30 June 2021 at the Department of Radiology and Nuclear Medicine, University Hospital Basel, Switzerland, with the following steps (Figure 1): First, all patients who underwent MRI using a 1.5T scanner (Siemens MAGNETOM Avanto FIT 1.5T, Siemens Healthcare, Erlangen, Germany) as part of the diagnostic stroke workup for suspected stroke or transient ischemic attack (TIA) were preselected. Afterwards, MRI scans were reviewed for completeness and quality and regarding diagnosis of ischemic stroke or other acute pathologies. If no other acute pathology was detected, consent was obtained and patients were additionally examined using a new prototype of 0.55T scanner (Siemens MAGNETOM FreeMax 0.55T, Siemens Healthcare, Erlangen, Germany) directly after the 1.5T examination. If 0.55T datasets were complete and of good quality the patient was included in the study. Included patients without ischemic stroke on 1.5T imaging or any other acute pathology were defined as the control group. Patients were excluded if they fulfilled the following exclusion criteria:(a)Incomplete dataset 0.55T or 1.5T examination(b)Insufficient image quality 0.55T or 1.5T examination(c)No stroke but other acute pathology within initial 1.5T scan(d)No informed consent for additional 0.55T examination(e)Too large time difference between 1.5T and 0.55T examination (cut off 2 h)(f)Foreign materials not authorized for 0.55T scanners (i.e., cardiac pacemakers)
Figure 1Workflow of patient selection and data acquisition.
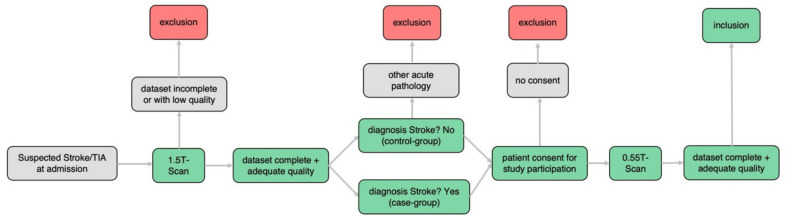


The 1.5T scanning protocol was in accordance with the hospital’s internal standard protocol for emergency stroke diagnostics including axial DWI/ADC, FLAIR, and SWI (susceptibility-weighted imaging) sequences (Table 1). The 0.55T protocol (Table 1) corresponded to the standard protocol regarding the sequences used and was maximally adapted to the 1.5T protocol as far as technically possible (same slice thickness (ST) and slice spacing (SP); comparable in-plane resolution) to ensure the most sufficient scanner comparison. After subsequent verification with respect to data completeness (scan protocols with complete image acquisition) and image quality (artifacts, image contrast), the datasets were transferred to the Picture Archiving and Communication System (PACS, General Electric (GE), Boston, Massachusetts, MA, USA) for further analysis.

### 2.2. Data Analysis

Data analysis was performed in a three-step procedure (Figure 2): First, the 0.55 T and 1.5 T datasets were evaluated using Likert rating. Second, a reading study was performed regarding identification and localization of DWI and FLAIR lesions. Finally, a correlation analysis of DWI and FLAIR lesion volumes was performed.

#### 2.2.1. Likert Rating

Likert rating was performed by a neuroradiologist and a neuroradiologist in training with experience of 9 and 5 years. Each acquired 0.55 T or 1.5 T DWI/ADC and FLAIR dataset was rated with respect to the following criteria with a numerical value between 1 and 10 (1 maximum poor, 10 maximum good):(a)Overall image quality;(b)Resolution;(c)Noise;(d)Contrast;(e)Diagnostic quality.

Sample DWI/ADC and FLAIR sequences from a 3T scanner (Siemens MAGNETOM Skyra 3T, Siemens Healthcare, Erlangen, Germany) were set as gold standard (numerical value = 10). Dataset assessment was PACS-based using a standardized bookmark. Both raters were blinded to the results of the other rater.

#### 2.2.2. Reading Study

Reading of 0.55 T and 1.5 T datasets was performed PACS-based and blinded (no clinical information, no image information) by two neuroradiologists with 8 and 13 years of professional experience, the latter defined as the most experienced rater. PACS-based post-processing as part of image analysis was allowed. For each dataset, the following reading tasks were conducted:(a)Evaluation stroke yes/no;(b)Number of DWI lesions: 0, 1, 2–10, >10;(c)DWI lesion main localization (especially in the case of multiple DWI lesions);(d)Number of FLAIR lesions: 0, 1, 2–10, >10;(e)FLAIR lesion main localization (especially in the case of multiple FLAIR lesions).

Reading was performed for only 24 of the 27 patient datasets (Figure 2, Table 2) because the time gap between the 1.5 T and 0.55 T examination was >1 h in three cases and possible data distortions caused by this time delay (e.g., infarct demarcation increasing in the meantime and therefore easier detectability of the lesions) should be excluded.

The clinical neuroradiological report and final neurological diagnosis (Table 2) were defined as underlying gold standard for the accuracy of the reading study.

#### 2.2.3. Segmentation of DWI/ADC and FLAIR Lesions

For segmentation, the 0.55 T and 1.5 T datasets were transferred to a research server and post-processing software (NORA Medical Imaging Platform Project, University Medical Center Freiburg, Freiburg, Germany). Then, segmentation of stroke lesions was performed separately in each of the 0.55 T and 1.5 T DWI, ADC, and FLAIR sequences by two radiologists with experience of 4 and 5 years. Stroke lesions with multifocal and inhomogeneous confluent distribution or punctate configuration (too small for segmentation) were excluded for the segmentation process because in these cases sufficient segmentation was limited, and the bias of the results possibly caused by this should be avoided (Figure 2). In total, segmentation was performed for six 0.55 T and 1.5 T datasets.

### 2.3. Statistical Analysis

For statistical evaluation of Likert rating, a mean of the ratings of readers 1 and 2 was first calculated for each 0.55 T and 1.5 T patient dataset and evaluation point (a)–(e). Subsequently, a Wilcoxon signed rank test was used to evaluate significant or non-significant differences in Likert rating between the 0.55 T and 1.5 T sequences. Then, inter-reader comparisons were performed to determine intraclass correlation coefficients (ICC).

Calculation of sensitivity and specificity of readers 1 and 2 in the reading study was performed in relation to the gold standard.

Volume correlation of infarct volume between the 0.55 T and 1.5 T DWI and ADC sequences and volume correlation of infarct demarcation between the 0.55 T and 1.5 T FLAIR sequences were performed by first obtaining a mean volume for each of the six 0.55 T and 1.5 T DWI, ADC, and FLAIR datasets from readers 1 and 2. Afterwards, the Wilcoxon signed rank test was used to examine significant differences in infarct volume and volume of infarct demarcation between the 0.55 T and 1.5 T datasets (*p* < 0.05). In addition, an inter-reader correlation was performed for the segmented volumes of the DWI, ADC, and FLAIR datasets by calculating an intraclass correlation coefficient (ICC).

## 3. Results

A total of 27 complete and artifact-free datasets (17 stroke cohort; 10 control group) were acquired (mean age ± standard deviation, 71 years ± 15; 12 women [44%]).

Most included patients had mild neurologic symptoms (for details see Table 2) at admission and a low NIHSS score (1.88 ± 2.52).

### 3.1. Likert Rating

#### 3.1.1. DWI/ADC Datasets

Regarding overall image quality (a), resolution (b), contrast (d), and diagnostic quality (e), average Likert ratings of readers 1 and 2 (Figure 3) were significantly better for the 1.5 T sequences than the 0.55 T sequences: (a) *p* < 0.001; (b) *p* < 0.001; (d) *p* = 0.001; (e) *p* < 0.001. Regarding noise (c) 0.55 T sequences were significantly superior (*p* < 0.026) to the 1.5 T sequences. Inter-reader comparisons showed high levels of agreement between readers 1 and 2 (ICC: (a) 0.77; (b) 0.78; (c) 0.84; (d) 0.71; (e) 0.88).

#### 3.1.2. FLAIR Datasets

Regarding overall image quality (a), noise (c), and diagnostic quality (e), average Likert ratings of both readers (Figure 4) were significantly better for the 1.5 T sequences than the 0.55 T sequences: (a) *p* < 0.0027; (c) *p* < 0.001; (e) *p* < 0.0292). There was no significant difference for the criteria resolution (b) and contrast (d) ((b) *p* = 0.1039 and (d) *p* = 0.7890).

In the inter-reader comparison, there was medium to high agreement (ICC: (a) 0.64; (b) 0.87; (c) 0.64; (d) 0.73; (e) 0.71).

### 3.2. Reading Study

The reading study was performed with a total of 24 of the initially acquired 27 patient datasets (14 stroke cohort and 10 control group; mean age ± standard deviation, 74 years ± 14; 46% women) because in three datasets from the stroke cohort the time gap between the 1.5 T scan and 0.55 T scan was >1 h and possible result distortions caused by this should be excluded (Table 2).

The average time gap between the 1.5 T scans and 0.55 T scans was 36.8 ± 14.7 min.

#### 3.2.1. DWI/ADC Datasets

There were no false positive findings in the 0.55 T sequences by readers 1 and 2, meaning no lesions were detected in the control group datasets (specificity at 0.55 T of readers 1 and 2: 100%). Reader 1 failed to detect a stroke in one case in both the 0.55 T and 1.5 T datasets (sensitivity of reader 1 at 0.55 T compared to gold standard: 92.9%).

Reader 2 did not detect two DWI lesions in the 0.55 T datasets but detected all lesions in the 1.5 T datasets (sensitivity of reader 2 at 0.55 T compared to gold standard: 85.6%). Both missed lesions had a point-like pattern, one being located subcortical/cortical and one infratentorial (Figure 5 and Figure 6). Regarding the number of stroke lesions and lesion localization, there was complete agreement between readers 1 and 2 at the 1.5 T and 0.55 T datasets. Assessment of the extent of stroke as well as safe anatomic stroke localization was thus equivalent in the 0.55 T sequences compared with the 1.5 T sequences (Figure 7).

#### 3.2.2. FLAIR Datasets

In only one case did an inter-reader difference occur with respect to the assessment for a singular pontine lesion detected by reader 1 and not detected by reader 2. In the remaining 23 0.55 T and 1.5 T datasets, no inter-reader differences occurred. Reader 1 failed to detect single (1–2) FLAIR lesions with localization in the corona radiata in the 0.55 T FLAIR sequences compared with the 1.5 T FLAIR sequences in three cases. In contrast, there was complete agreement in the remaining 21 datasets. Reader 2 failed to detect single (1–2) FLAIR lesions with localization in the corona radiata in two cases in the 0.55 T FLAIR sequences compared with the 1.5 T FLAIR sequences. These cases overlapped with the FLAIR lesions from reader 1 that were not detected in the 0.55 T sequences. In contrast, there was complete agreement in the remaining 22 datasets.

### 3.3. Segmentation of DWI/ADC and FLAIR Lesions

There was no significant difference regarding segmented lesion volumes between the 0.55 T and 1.5 T datasets (DWI *p* = 0.375, ADC *p* = 0.63, FLAIR *p* = 0.38). Inter-reader comparisons showed high levels of agreement for the volume segmentations of readers 1 and 2 for the individual DWI, ADC and FLAIR sequences (ICC DWI: 0.811, *p* = < 0.0014; ICC ADC: 0.89, *p* = < 0.0001; ICC FLAIR: 0.909, *p* < 0.0001).

## 4. Discussion

In our study, no significant differences were observed between the 0.55 T and 1.5 T sequences for FLAIR resolution and contrast regarding the Likert ratings. For DWI/ADC noise, the 0.55 T sequences were even significantly superior. For the diagnostic accuracy readings, the 0.55 T sequences were non-inferior for one of the two readers, meaning that DWI/ADC lesions were detected with the same specificity and sensitivity by the more experienced reader (100% and 92.9%, respectively). Regarding FLAIR lesions, there was almost complete agreement between the 0.55 T and 1.5 T sequences. In addition, we demonstrated that there were no significant differences between the 0.55 T and 1.5 T sequences in infarct volume and volume of infarct demarcation.

To the best of our knowledge, no comparable study has performed a 0.55 T versus 1.5 T scanner comparison regarding stroke imaging before. However, Mehdizade et al. [8] demonstrated, in a collective of a total of 18 patients, that an open low-field 0.23T scanner (Outlook, Marconi Medical Systems, Cleveland, OH, USA) can detect stroke lesions with equal confidence compared with a 1.5 T scanner (Eclipse, Marconi Medical Systems, Cleveland, OH, USA) using the acquired DWI sequences. In contrast, the diagnostic performance of the associated ADC maps was inferior for the 0.23T system. Terada et al. [9] investigated the same issue in a total of 24 patients with acute ischemic stroke using a 0.3T scanner (AIRIS II, Hitachi Medical Corporation, Tokyo, Japan) in comparison with a 1.5 T scanner (Gyroscan ACS NT, Philips Medical Systems, Hamburg, Germany). Similarly, the performance of the low-field MRI system was non-inferior, only more vulnerable to motion artifacts. The above-mentioned results are basically in agreement with the results of our study.

### 4.1. Likert Rating

The largest differences between DWI/ADC sequences at 0.55 T and 1.5 T were observed for resolution (6.2 ± 0.9 versus 7.6 ± 0.8) and contrast (7.6 ± 0.7 versus 6.0 ± 1.0). There may be several reasons for this: The resolution of the 0.55 T DWI/ADC sequences is nominally (about 14%) worse (pixel spacing 1.67 mm × 1.67 mm versus 1.44 mm × 1.44 mm; see Table 1). In addition, the bandwidth (BW) of the 0.55 T DWI/ADC sequences is significantly lower (BW 842 versus 1490, see Table 1). Therefore, the pass-through speed of the K space is slower compared to the 1.5 T DWI/ADC sequences. As a result, the data acquisition time on the 0.55 T scanner is longer and hence the time span of T2 * relaxation. According to the laws of the Fourier transformation, this is reflected in a signal reduction in K space and therefore a loss of resolution in spatial space (smoothing). The relatively high inferiority of the 0.55 T DWI/ADC sequences regarding contrast can be explained by the lower resolution (see above). This results in larger partial volume effects, which lead to lower contrast. Moreover, the 0.55 T scanner provides a priori a lower resonance signal (fewer protons are polarized) due to its lower magnetic field. The inferior head coil with only eight channels (1.5 T 32 channel head coil) also contributes to this effect. Interestingly, 0.55 T was superior to 1.5 T regarding DWI/ADC noise. Since the signal from the 0.55 T scanner is fundamentally inferior to the 1.5 T scanner due to the lower field strength and the inferior gradient and coil system, the latest post-processing applications for noise reduction were integrated into the 0.55 T system (deep resolve gain; Deep Resolve—Mobilizing the power of networks, Siemens Healthineers White paper, Behl et al.). This artificially reduces the visible and thus subjective noise noticeably without influencing or improving the signal and most likely explains the 0.55 T superiority regarding noise. With respect to FLAIR sequences, 0.55 T was non-inferior in resolution and contrast. This is because the resolution of the FLAIR sequences is already nominally better than the DWI/ADC 0.55 T sequences (Table 1) and the difference to the 1.5 T FLAIR sequences is therefore less detectable visually. In addition, the signal loss during K space acquisition is significantly lower for the FLAIR sequences than for the single-shot epi sequences (DWI/ADC), inducing less smoothing and not significantly reducing contrast. However, the decisive criterion for the radiologist is the diagnostic quality of the sequences. In this regard, there were only minor differences between the 0.55 T and 1.5 T DWI/ADC and FLAIR sequences (Figure 3 and Figure 4) to the disadvantage of the 0.55 T system. Nevertheless, the advantages of the low-field MRI are lower costs for installation and maintenance (lower weight, smaller device size, no quench pipe, smaller MRI cabins) and thus for image acquisition, which in turn could increase future availability in developed economies, but especially in undeveloped areas [10,11].

#### Reading Study

Although the 0.55 T DWI/ADC sequences were maximally optimized (variation of b values, resolution and scan duration, application of artificial intelligence-based algorithms to increase visible resolution, as well as noise reduction), acute DWI lesions with punctuated configuration and cortical/subcortical or infratentorial localization could not be reliably detected in individual cases. In our opinion, possible causes are a too low contrast-to-noise ratio (CNR) and signal-to-noise ratio (SNR) or a too low resolution of the 0.55 T sequences. In conclusion, a too weak signal is the most likely cause. Future developments in coil design and the usability of techniques such as simultaneous multi-slice acquisition or stronger gradient systems may improve the SNR available per unit time and possibly the detection of small lesions.

In principle, the measurement time could simply be extended to generate more signal. However, this would lead to an increase in motion artifacts and long measurement times are counterproductive, especially in stroke patients, so a tradeoff between acquisition time and signal must be found. Overall, the impact of non-detection of a stroke lesion on further patient care is crucial. From a meta-analysis of 12 studies by Edlow et al. [12], it is known that, regardless of field strength, the prevalence of DWI negative acute ischemic strokes is approximately 6.8%. Factors associated with false negative DWI include stroke lesions of the posterior circulation, small volume, in which MRI was performed within the first 6 h of onset, or the NIHSS was <4 [12,13]. Some of these factors also apply to DWI lesions not detected in our study. Confirmed data on morbidity and mortality due to non-detection of stroke lesions in this context are not available. However, it is known that patients with missed strokes have a higher risk of recurrent strokes because of failure of stroke evaluation or secondary stroke prevention [13,14]. Related to the results of our study, we therefore recommend triage of patients with suspected stroke or TIA in the emergency setting: patients with mild, non-specific neurological symptoms or compatible with a posterior circulation stroke should thus be primarily examined on 1.5 T–3T devices to reliably detect even the smallest DWI lesions. Patients with clear or severe neurological symptoms, in the context of a diagnosis of exclusion or status after stroke for the evaluation of infarct demarcation, infarct size or complications after lysis (e.g., hemorrhage), could be examined sufficiently, safely, and equally on the 0.55 T system. This is also underlined by the results of our segmentation analysis, in which no significant differences in infarct volumes and volumes of infarct demarcation could be seen between the 0.55 T and 1.5 T sequences.

### 4.2. Limitations

There are several limitations that need to be addressed: First, a 1.5 T device of routine clinical use is the gold standard in this study. The question arises whether, for example, more lesions would have been detected at 3T or at 7T. Second, both scanners differ in respect to their gradient and coil system as well as the field strength. Therefore, it ultimately remains unclear whether the reason for a lack of delineation of the smallest lesions at 0.55 T is due to the field strength, the weaker gradient system, or even technical specifications such as the coils used. Third, the study cohort—while prospective—is still relatively small. Larger-scale studies to further define indications for stroke imaging at 0.55 T are needed and should assess whether scanner choice has an impact on patient outcomes. Fourth, the patients included in the study (control group and stroke cohort) were a convenient sample with limited representativeness in relation to the overall population (patients with suspected stroke or TIA). However, from our point of view this limitation plays only a minor role, because our study was primarily concerned with the basic evaluation of low-field MRI imaging in the context of a scanner comparison and less with the definition of, for example, possible selection criteria in the context of MRI-based stroke diagnosis, for example.

Fifth, there may be a lack of external validity, as the interpretation of the 0.55 T sequences may be difficult, especially for inexperienced radiologists. This may be aggravated in regions with low MRI availability and therefore a lack of neuroradiological specialists. On the other hand, this study presents an initial experience with 0.55 T MRI stroke imaging and could serve as a help for the interpretation.

In conclusion, low-field MRI stroke imaging at 0.55 T may not be inferior to scanners with higher field strength and thus has great potential as a low-cost alternative in future stroke diagnostics. However, there are limitations in the detection of very small infarcts. Further technical developments with follow-up studies must show whether these questions can be solved.

## Figures and Tables

**Figure 2 jcm-11-02798-f002:**
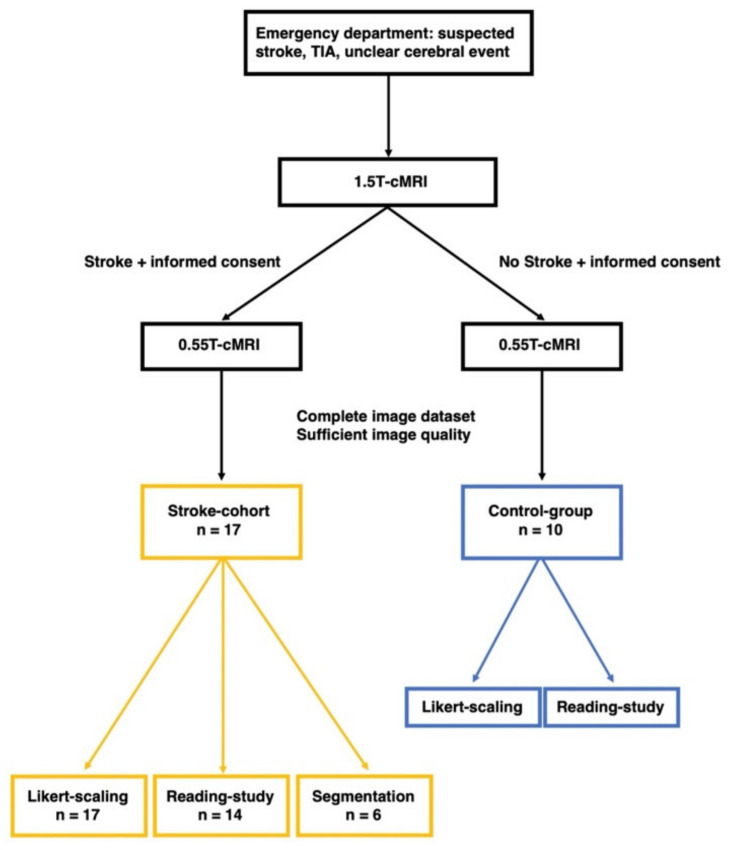
Workflow of data processing.

**Figure 3 jcm-11-02798-f003:**
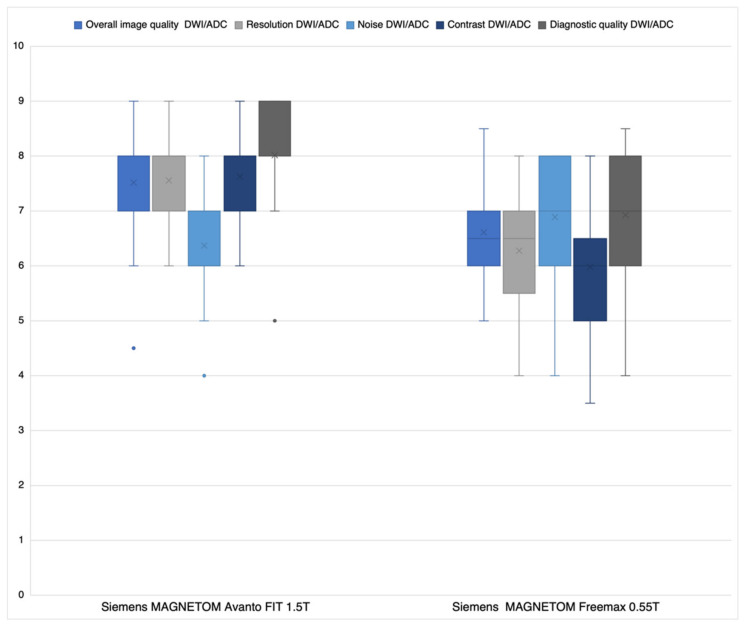
Average Likert-scoring reader 1 and 2 DWI/ADC sequences.

**Figure 4 jcm-11-02798-f004:**
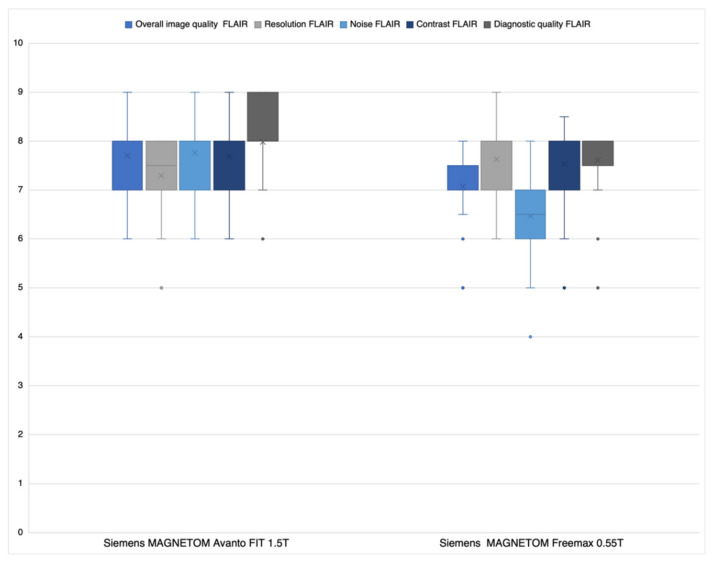
Average Likert-scoring reader 1 and 2 FLAIR sequences.

**Figure 5 jcm-11-02798-f005:**
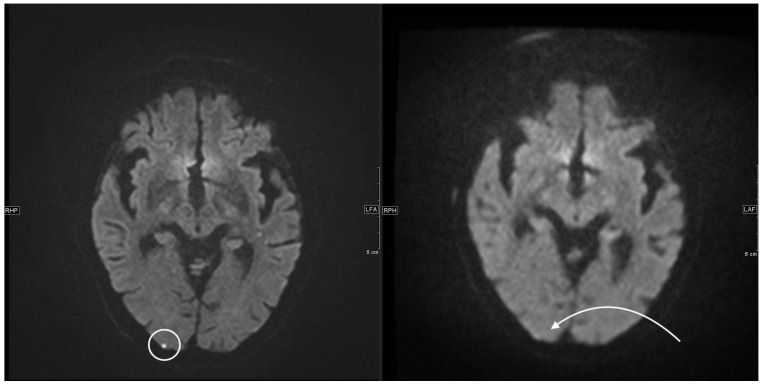
Left side, axial 1.5 T DWI sequence with sharply hyperintense delineable, punctiform DWI lesion located cortico-subcortical occipital right. On the right side, corresponding 0.55 T axial DWI sequence with the same slice localization without sufficiently detectable lesion.

**Figure 6 jcm-11-02798-f006:**
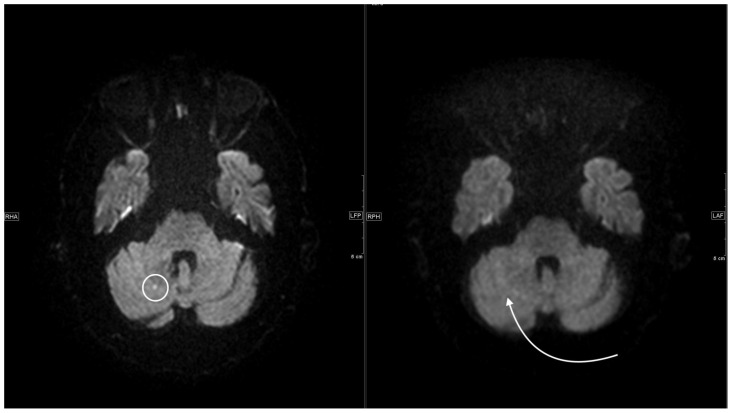
Left side, axial 1.5 T DWI sequence with sharply hyperintense delineable, punctiform DWI lesion located cerebellar right. On the right side, corresponding 0.55 T axial DWI sequence with the same slice localization without sufficiently detectable lesion.

**Figure 7 jcm-11-02798-f007:**
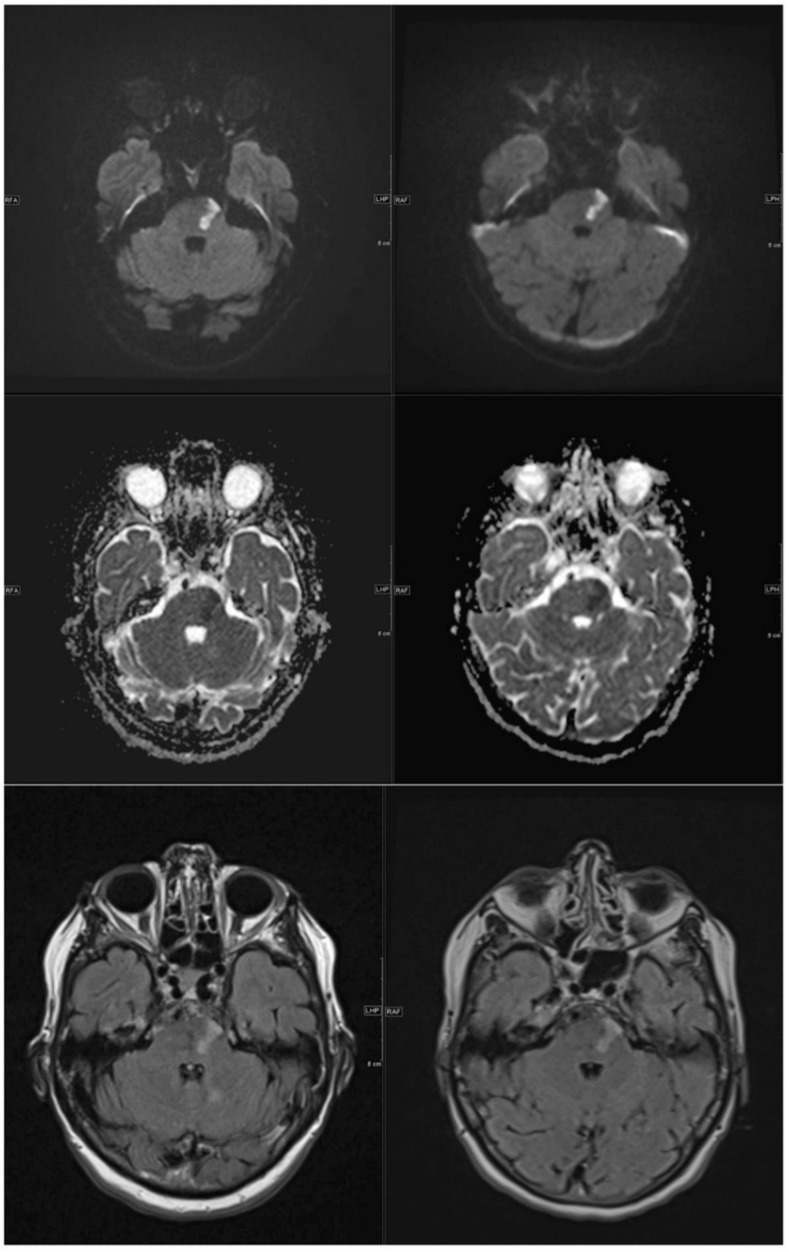
Left images from top to bottom axial 1.5 T DWI, ADC and FLAIR sequence with sharply delineated, demarcated, subacute pons infarct on the left. Right side, corresponding axial 0.55 T DWI, ADC and FLAIR sequence at the same slice localization with equivalent infarct morphology and thus identical diagnostic quality.

**Table 1 jcm-11-02798-t001:** Scan protocols, Siemens MAGNETOM Avanto FIT 1.5 T and Siemens MAGNETOM FreeMax 0.55 T.

	Siemens MAGNETOM FreeMax 0.55 T	Siemens MAGNETOM Avanto Fit 1.5 T
**FLAIR (fluid attenuated inversion recovery) transversal**		
Field strength in T	0.55	1.5
FOV (field of view) in mm^2^	209 × 230	187 × 230
ST (slice thickness) in mm	3	3
SS (slice spacing)	3.6	3.6
Number of slices	40	40
PS (pixel spacing) in mm^2^	1.28 × 1.03	0.9 × 0.9
TR (repetition time) in msec	7780	8510
TE (echo time) in msec	96	112
TI (inversion delay) in msec	2368.8	2460
Turbo Factor	15	19
TA (time of acquisition) in min	5:28	3:26
BW (Bandwidth)	150	130
**3D SWI (susceptibility weighted imaging) transversal**		
Field strength in T	0.55	1.5
Sequence type	Multi shot 3D EPI	3D FLASH
FOV (field of view) in mm^2^	201 × 230	194 × 230
ST (slice thickness) in mm	3	3
Number of slices	40	48
PS (pixel spacing) in mm^2^	0.94 × 0.8	1.12 × 0.9
TR (repetition time) in msec	172	48
TE (echo time) in msec	100	40
Parallel imaging	-	GRAPPA factor 2
TA (time of acquisition) in min	2:23	2:17
BW (Bandwidth)	276	80
**Single shot diffusion EPI (echo-planar imaging) transversal**		
Field strength in T	0.55	1.5
FOV (field of view) in mm^2^	220 × 220	230 × 230
ST (slice thickness) in mm	3	3
SS (slice spacing)	3.6	3.6
Number of slices	40	40
PS (pixel spacing) in mm^2^	1.67 × 1.67	1.44 × 1.44
b values in s/mm^2^	0, 1000	0, 1000
TR (repetition time) in msec	7400	6200
TE (echo time) in msec	102	103
Parallel imaging	GRAPPA factor 2	GRAPPA factor 2
TA (time of acquisition) in min	4:35	2:04
BW (Bandwidth)	842	1490

**Table 2 jcm-11-02798-t002:** Detailed patient data.

Patient	Patient Age	Neurological Symptoms at Admission	NIHSS	Control-Group (C), Stroke-Cohort (S)	Final Radiological Report	Time Gap between Scans in min	Time Gap between Onset and Scan 1 in min
Patient 1	87	facial droop, dysarthria, hemiparesis right side	3	S	acute to subacute infarct corpus nuclei caudati and cranial parts of the nucleus lentiformis left side	46	1166
Patient 2	88	ataxic gait and standing	no data	S	acute to subacute infarct lateral pontin left side	37	unknown
Patient 3	82	visual deficit	no data	S	acute to subacute punctiform infarcts parietal left and cerebellar right	93	unknown
Patient 4	84	intermittent dysarthria	1	S	acute to subacute infarct thalamus left side	33	unknown
Patient 5	58	facial droop, hemiparesis right side	10	S	multiple subacute infarcts posterior circulation on both sides	40	1050
Patient 6	65	low-grade facial paresis left side	1	S	acute to subacute infarcts of the thalamus and occipital/occipitotemporal right side	20	1175
Patient 7	65	dysdiadochokinesis right side	no data	S	subacute punctiform infarcts frontal and parietal left side	24	1704
Patient 8	75	facial droop right side	0	S	punctiform infarct gyrus postcentralis left side	22	1135
Patient 9	82	confusion	no data	S	acute to subacute infarcts frontal and parietal left side	32	unknown
Patient 10	79	hemiataxia left side	2	S	acute to subacute infarct thalamus right side	42	1492
Patient 11	86	dysarthria, ataxia right leg	3	S	acute infarct posterolateral pons left side	25	unknown
Patient 12	83	leg-emphasized hemiparesis left side	5	S	subacute infarct gyrus precentralis right side	31	unknown
Patient 13	89	visual deficit	2	S	acute cortical infarcts parietooccipital and frontal right side	38	2198
Patient 14	69	no data	no data	S	subacute infarct central left side	42	unknown
Patient 15	73	amaurosis fugax	0	C	no stroke	25	1197
Patient 16	29	strong nystagmus to left side, headache right frontal, vertigo	no data	C	no stroke	48	unknown
Patient 17	70	atypical transient global amnesia	no data	C	no stroke	44	unknown
Patient 18	87	aphasia	no data	C	no stroke	32	unknown
Patient 19	74	vertigo	0	C	no stroke	25	826
Patient 20	60	aphasia	0	C	no stroke	21	unknown
Patient 21	44	vertigo	0	C	no stroke	49	unknown
Patient 22	80	short-term loss of vision left side	no data	C	no stroke	35	2092
Patient 23	84	vertigo, gait instability	no data	C	no stroke	48	425
Patient 24	84	vertigo, gait instability	no data	C	no stroke	32	unknown
Patient 25	53	aphasia	1	excluded	subacute punctiform infarcts frontal and parietal left side	916	1197
Patient 26	59	transient global amnesia	0	excluded	bilateral punctiform diffusion-restrictions of the hippocampus head	2936	1394
Patient 27	46	facial droop, descending arm left side	2	excluded	acute to subacute infarct lenticostriatal right side	2812	1406

## Data Availability

The data presented in this study are available on request from the corresponding author.

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
