# Peer review of "Potential of Stroke Imaging Using a New Prototype of Low-Field MRI: A Prospective Direct 0.55 T/1.5 T Scanner Comparison"

_jcm, 2022, doi:10.3390/jcm11102798_

Round 1

Reviewer 1 Report

The paper is interesting, innovative and well processed by a small number of respondents, which the authors stated.
Currently, there are two trends in MR imaging, on the one hand more and more 1.5 T MR scanners are being replaced by 3T and 7T devices, while on the other hand in developing countries, due to low prices and therefore easier availability thay use low field MRI of 0.5 T. Clinically, the basis of MR imaging lies in the contrast of tissues, which cannot be replaced by other modalities, such as CT. For that reason, the strength of the static magnetic field and the price are of great importance. As the authors stated, the problem exists with the detection of small infarcts and with further development in the future, they may become more widespread and available in stroke units.

Author Response

Dear Reviewer,

On behalf of everyone involved in the manuscript, I would like to thank you very much for the quick and positive review.

Within the scope of a few minor criticisms by the second reviewer, a couple of sections of the final manuscript have been changed and supplemented.

With kind regards

Thilo Rusche

Reviewer 2 Report

Overall, the manuscript is interesting and well-written. Nevertheless, there are major issues that need to be clarified. 

It is not clear how the control group was selected. The authors must explain it better. Was there a match between the characteristics of cases and controls?

Did the authors define any exclusion criteria?

Did the author calculate the sample size?

The authors must mention exactly what were the inclusion criteria and the time frame in which subjects were recruited. Two months seem too much time to recruit only 27 CONSECUTIVE patients.

The authors should report the average time between the realization of 1.5T MRI and 0.55T MRI.

It would be important to understand the clinical characteristics of the patients from whom the images were obtained. Moreover, the stroke characteristics are also important. They are part of the pre-test probability that definitely influences the diagnosis of stroke. The time from symptoms’ onset to MRI would also be important.

In parallel, do the authors have some data about the radiological characteristics of all included cases? I would recommend the authors provide a Table summarizing the neuroradiological findings of the subjects that allowed the diagnosis of ischemic stroke.

The authors must include the choice of using a convenience sample as a limitation of the study.

Countries and regions with a lack of MRI availability for acute ischemic stroke may also have a lack of neuroradiologists. The authors present Fig. 4 and 5 by which one can imagine that the accurate interpretation of the 0.55T sequences would require more experience in the field. In this situation, using 0.55T MRI may pose a risk if the radiologist is less experienced. Therefore, the results of this study may lack from external validity. The authors must discuss this issue.

“In conclusion, low-field MRI stroke-imaging at 0.55T is non-inferior to scanners with higher field-strength and thus has great potential as a low-cost alternative in future stroke diagnostics” – This study does not have enough power to conclude that 0.55T is non-inferior to scanners with field-strength.  The results just suggest this that there may be a non-inferiority.  

Author Response

Dear Reviewer,

Thank you for the detailed and quick review and constructive criticism.

Within the attached Word-document you will find a point-by-point response to the relevant questions and criticisms.  

In addition, a correction and addition to the final manuscript was made on this basis.

Yours sincerely

Thilo Rusche

Reviewer 3 Report

The comparison between 2 types of MRI scan, one with low level of exposure, is very interesting and represent the main contribution of this article compared with other ones. Well written with text very easy to read and understanding. Nice discussions and clear conclusions for the readers.

Author Response

(The authors gave the same response as above.)

Round 2

Reviewer 2 Report

The article is now suitable for publication.